# Improving the Flexural Response of Timber Beams Using Externally Bonded Carbon Fiber-Reinforced Polymer (CFRP) Sheets

**DOI:** 10.3390/ma17020321

**Published:** 2024-01-08

**Authors:** Walid Mansour, Weiwen Li, Peng Wang, Cheikh Makhfouss Fame, Lik-ho Tam, Yao Lu, Md. Habibur Rahman Sobuz, Noha Yehia Elwakkad

**Affiliations:** 1Guangdong Provincial Key Laboratory of Durability for Marine Civil Engineering, Shenzhen University, Shenzhen 518060, China; wangpeng@ust.hk (P.W.); fame@szu.edu.cn (C.M.F.); luyao2020@email.szu.edu.cn (Y.L.); 2Civil Engineering Department, Faculty of Engineering, Kafrelsheikh University, Kafrelsheikh 33516, Egypt; 3Department of Civil and Environmental Engineering, The Hong Kong University of Science and Technology, Hong Kong 999077, China; 4School of Transportation Science and Engineering, Beihang University, Beijing 100191, China; leo_tam@buaa.edu.cn; 5Department of Building Engineering and Construction Management, Khulna University of Engineering & Technology, Khulna 9203, Bangladesh; habib@becm.kuet.ac.bd; 6Civil Engineering Department, Faculty of Engineering, Damanhour University, Damanhour 22511, Egypt; amatallah_n@hotmail.com

**Keywords:** timber, strengthening, carbon-fiber-reinforced polymer (CFRP), finite element analysis (FEA), flexural rigidity, ductility index

## Abstract

This paper presents a numerical investigation of the flexural behavior of timber beams externally strengthened with carbon-fiber-reinforced polymer (CFRP) sheets. At first, the accuracy of linear elastic and elastic-plastic models in predicting the behavior of bare timber beams was compared. Then, two modeling approaches (i.e., the perfect bond method and progressive damage technique using the cohesive zone model (CZM)) were considered to simulate the interfacial behavior between FRP and timber. The models were validated against published experimental data, and the most accurate numerical procedure was identified and subsequently used for a parametric study. The length of FRP sheets varied from 50% to 100% of the total length of the beam, while different FRP layers were considered. Moreover, the effects of two strengthening configurations (i.e., FRP attached in the tensile zone only and in both the tensile and compressive zones) on load-deflection response, flexural strength, and flexural rigidity were considered. The results showed that elastic-plastic models are more accurate than linear elastic models in predicting the flexural strength and failure patterns of bare timber beams. In addition, with increasing FRP length, the increase in flexural strength ranged from 10.3% to 52.9%, while no further increase in flexural strength could be achieved beyond an effective length of 80% of the total length of the beam. Attaching the FRP to both the tensile and compressive zone was more effective in enhancing the flexural properties of the timber beam than attaching the FRP to the tensile zone only.

## 1. Introduction

Over the past decades, fiber-reinforced polymer (FRP) has been increasingly utilized in aerospace, automotive, marine, and civil engineering due to its excellent mechanical and physical characteristics, such as advanced corrosion resistance, light weight, versatility, high strength- and stiffness-to-weight ratio (CFRP in particular), and so on [1,2,3,4]. These outstanding properties are particularly desirable for the strengthening and retrofitting of civil construction made from conventional materials such as concrete, steel, and timber. This is particularly true in wood engineering where timber as a stand-alone construction material exhibits durability issues owing to the increasing need for a longer service life of civil infrastructures under harsh environmental conditions. In this sense, FRP materials, especially for CFRP, have found their way into wood engineering as a strengthening or retrofitting material for primary structural elements such as beams and columns resulting in hybrid composite structures with enhanced strength, stiffness, and ductility.

To ensure the optimum strengthening of timber beams, the load should be transferred efficiently from the timber substrate to the FRP layers. This implies that suitable connection methods are required to ensure the structural efficiency of the hybrid timber structures. Adhesive bonding is a natural choice for assembling timber and FRP materials and has been increasingly used in civil engineering as a common connection approach for structural members due to several competitive benefits such as low manufacturing costs, ability to assemble dissimilar materials, relatively uniform stress distribution, and ease of installation [5,6,7,8,9,10].

When subjected to bending loads, timber beams are relatively weak on the bottom side where high tensile stresses occur. Given this, FRP sheets are often adhesively bonded to the soffit of the timber beams to increase flexural stiffness, strength, ductility, and energy absorption as well as prevent premature failure of timber beams [11,12,13,14,15]. In this case, the FRP–timber bonded structures can exhibit various failure modes and damage mechanisms, which inherently depend upon several parameters, such as the timber strength, FRP sheets strength, bulk adhesive strength, adhesive–timber interface strength, and adhesive–FRP interface strength. These failure modes include timber cohesive cracking due to flexural stress, FRP rupture or delamination, FRP–timber interface debonding, or a combination of these. As a result, the prediction of the failure mechanism and flexural capacity of timber–FRP bonded structures is very challenging due to the interaction of several damage mechanisms.

FE simulations may be a desirable choice to effectively predict the flexural response and failure mechanisms of timber beams strengthened with FRP considering different configuration conditions. Some studies [13] have modeled the timber beams as an orthotropic linear elastic material without considering their non-linear behavior (i.e., the maximum stress (or strain) is used as the failure criterion for timber and FRP rupture), while others [14,15,16,17,18] have assumed an elastic-perfect-plastic approach using the anisotropic Hill yield criterion [19] to describe the failure of the timber material. The reason is that although timber beams are prone to brittle failure in the tensile region under bending loads, their load-deflection curves usually exhibit a plastic behavior before failure [20]. The interfacial behavior between timber substrate and FRP has been simulated by either the perfect bond model [13,18] or the cohesive zone model (CZM) [14]. The perfect bond approach assumes that the adhesive strength exceeds by far the cohesion strength of timber and that interfacial failure between FRP and timber substrate will not occur, implying that the predicted failure path is limited to the timber and FRP materials. In contrast, the cohesive zone model (CZM) enables progressive damage, allowing slip and separation between two initially bonded surfaces. It can successfully mimic progressive interfacial debonding between the timber and FRP layers observed in the physical testing of timber beams strengthened with FRP.

Several influencing parameters may significantly affect the flexural behavior of timber beams strengthened with FRP. Kim and Harries [13], for instance, analyzed the effects of various CFRP and timber properties on the flexural response of the strengthened timber beams using three-dimensional (3D) finite element (FE) simulations. They found that there was an effective CFRP reinforcement ratio beyond which no further increase in strength was noticeable. They also claimed that the elastic modulus of the CFRP reinforcement had no obvious influence on the strength of the strengthened beam. In contrast, the mechanical properties of the timber could significantly affect the flexural response of the strengthened beams. In the research work of Shekarchi et al. [20], different strengthening configurations (including flat, U-shaped, and L-shaped pultruded glass FRP (GFRP) profiles) were considered, and their flexural performances in three-point bending tests were compared. The effects of the position of the GFRP (i.e., attached to the tensile or both tensile and compressive regions of the timber beams) on the flexural behavior of the strengthened beams were also evaluated. However, the effects of GFRP length and thickness (i.e., number of CFRP sheets) were not investigated. Test results showed an increase by 59%, 61%, 79%, and 209% in flexural rigidity, modulus of rupture, ductility, and energy absorption, respectively, of the strengthened timber beams with respect to bare timber beams. On the other hand, beams strengthened with flat GFRP experienced premature failure in comparison to those strengthened with U-shaped and L-shaped GFRP, indicating that the latter configurations showed greater flexural performance than the former. Recently, Harrach and Rad [14] compared the flexural response of timber beams that were externally or internally strengthened with CFRP. The study also investigated the impact of altering the placement of internal CFRP reinforcement along the beam height on the flexural behavior of the strengthened beam. Results demonstrated that attaching the CFRP externally to the timber beams was more effective than incorporating it within the timber layers.

From the aforementioned studies, it can be seen that different modeling approaches have been used to simulate the behavior of FRP–timber members. There are also several viewpoints on how to simulate the interfacial behavior between FRP and timber. However, a systematic analysis comparing the efficiency of those techniques in modeling the flexural behavior (i.e., failure mode, damage mechanisms, and flexural capacity) of the strengthened timber beams is yet to be performed. It is crucial to figure out the fundamental failure mechanism of hybrid structures made from different materials. However, scarce studies can be found to address the interaction issue of the different failure modes of FRP–timber members. On the other hand, several parameters, including the length, thickness, and position of the strengthening FRP sheets, may affect the performance of the strengthened timber beam under bending loads. However, limited research has been conducted to address these issues.

To overcome these research gaps, this work thoroughly investigates the flexural behavior of timber beams strengthened with FRP under quasi-static four-point bending using three-dimensional finite element analysis. At first, a comparative study was conducted to evaluate the accuracy of four modeling approaches: (i) timber modeled as linear elastic material; (ii) timber modeled as elastic-perfect-plastic material; (iii) FRP–timber interface simulated using a perfect bond model; and (iv) FRP–timber interface simulated using a progressive damage model. The models were validated with published experimental data in terms of load-deflection curves, failure patterns, load-bearing capacity, and stress distribution profiles, and the most accurate modeling approach was identified. Then, comprehensive parametric studies were performed on parameters that may influence the beams’ flexural performance. These parameters included (i) the FRP overlapping distance along the length of the timber beams; (ii) the FRP placement (i.e., attached to the tension zone or both tension and compressive zones); and (iii) the FRP thickness (number of layers). Results were reported in terms of failure modes, load-deflection responses, load-bearing capacity, flexural rigidity, and ductility index. The numerical procedure proposed in this work is expected to promote the incorporation of virtual testing for the design of hybrid FRP–timber structures in a cost-effective and labor-saving way so as to reduce to a minimum the expensive experimental testing prescribed in current design guidelines.

## 2. Finite Element Analysis

The commercial software ABAQUS 6.17 [21] was used to create a three-dimensional (3D) numerical model capable of representing the flexural behavior of the externally strengthened timber beams with CFRP layers. Numerous numerical models for the material of the timber beams are now available in the literature (i.e., elastic model as well as elastic-plastic model), and there are several viewpoints on how to simulate the cohesive surface between the CFRP layers and the timber substrate, such as the perfect bond model and cohesive surface model. The numerical investigation carried out during the verification process of the finite element model (FEM) compares and evaluates the results of the proposed four models with different assumptions of the interfacial surface between the timber substrate and CFRP layers, for the purpose of selecting the optimum FEM that can be successfully used in the parametric analysis.

### 2.1. Material Modeling

#### 2.1.1. Timber

Two different models were defined in this study to simulate the timber material: the first was an elastic model, while the second was an elastic-plastic model.

##### Elastic Model

Linear elasticity of the timber material is identified by providing the engineering constant parameters: the three moduli of elasticity $E_{1},\;E_{2},\;E_{3}$; Poisson’s ratios $\upsilon_{12},\upsilon_{13},\upsilon_{23}$; and the shear moduli $G_{12},\;G_{13},\;{and\;G}_{23}$, corresponding to the material’s principal directions. The stress-strain law of elastic compliance is governed by Equation (1), while the employed engineering constants are shown in Table 1 as per the recommendations of [20].
(1)$$\left\{\begin{array}{c} \varepsilon_{11}\\ \varepsilon_{22}\\ \varepsilon_{33}\\ \gamma_{12}\\ \gamma_{13}\\ \gamma_{23} \end{array}\right\}=\left[\left.\begin{array}{cccccc} 1/E_{1} & -\upsilon_{21}/E_{2} & -\upsilon_{31}/E_{3} & 0 & 0 & 0\\ -\upsilon_{12}/E_{1} & 1/E_{2} & -\upsilon_{32}/E_{3} & 0 & 0 & 0\\ -\upsilon_{13}/E_{1} & -\upsilon_{23}/E_{2} & 1/E_{3} & 0 & 0 & 0\\ 0 & 0 & 0 & 1/{\;G}_{12} & 0 & 0\\ 0 & 0 & 0 & 0 & 1/{\;G}_{13} & 0\\ 0 & 0 & 0 & 0 & 0 & 1/{\;G}_{23} \end{array}\right]\right.\left\{\begin{array}{c} \sigma_{11}\\ \sigma_{22}\\ \sigma_{33}\\ \sigma_{12}\\ \sigma_{13}\\ \sigma_{23} \end{array}\right\}\;$$

##### Elastic-Plastic Model

With regard to the elastic-plastic model, the anisotropic Hill yield criterion model was developed to express the plasticity characteristics of the timber material because it is valuable in tri-axial stress analysis [19,22]. The suggested model is a classification of the Huber–Mises–Hencky hypothesis for anisotropic materials that establishes a link between the material strengths and the corresponding directions. The Hill yield stress value f(σ) in terms of rectangular Cartesian stress components can be described using Equation (2).
(2)$$f\left(\sigma \right)=\sqrt{F{\left(\sigma_{22}-\sigma_{33}\right)}^{2}+G{\left(\sigma_{33}-\sigma_{11}\right)}^{2}+H{\left(\sigma_{11}-\sigma_{22}\right)}^{2}+2L\sigma_{23}^{2}+2M\sigma_{31}^{2}+2N\sigma_{12}^{2}}<{{(\sigma }^{0})}^{2}$$
where F, G, H, L, M, and N are constants theoretically derived from Equation (3) to Equation (8), respectively, depending on the timber strength properties for the various orientations.
(3)$$F=\frac{1}{2}(\frac{1}{R_{22}^{2}}+\frac{1}{R_{33}^{2}}-\frac{1}{R_{11}^{2}})$$
(4)$$G=\frac{1}{2}(\frac{1}{R_{33}^{2}}+\frac{1}{R_{11}^{2}}-\frac{1}{R_{22}^{2}})$$
(5)$$H=\frac{1}{2}(\frac{1}{R_{11}^{2}}+\frac{1}{R_{22}^{2}}-\frac{1}{R_{33}^{2}})$$
(6)$$L=\frac{3}{2R_{23}^{2}}$$
(7)$$M=\frac{3}{2R_{13}^{2}}$$
(8)$$N=\frac{3}{2R_{12}^{2}}$$
where $\sigma_{11},\;\sigma_{22},{and\;\sigma }_{33}$ are the timber strength characteristics in the principal directions 1, 2, and 3, respectively; $\sigma_{12},\sigma_{31},{and\;\sigma }_{23}$ are the timber shear strength in the principal anisotropy planes 1-2, 3-1, and 2-3, respectively;$\;\sigma^{0}$ is the isotropic yield stress of the timber material; and $R_{11},\;{\;R}_{22},\;R_{33},\;R_{13},{\;R}_{12},{and\;R}_{23}$ are the potential plastic coefficients.

In the current model, it was imposed that the timber strengths were equal in directions 2 and 3 ($\sigma_{22}=\sigma_{33}$) [15]. In addition, the potential plastic coefficients, which were necessary to create the elastic-plastic model, were determined using Equations (9)–(13).
(9)$$\;\sigma^{0}=\sigma_{11}$$
(10)$$R_{11}=1.0$$
(11)$$R_{22}=R_{33}=\frac{\sigma_{22}}{\sigma_{11}}$$
(12)$$R_{23}=\frac{\sqrt{3}\sigma_{23}}{\sigma_{11}}$$
(13)$$R_{12}=R_{13}=\frac{\sqrt{3}\sigma_{12}}{\sigma_{11}}$$

The aforementioned Equations (10)–(13) are utilized to identify the potential plastic coefficients based on the timber mechanical strengths imported from the available numerical studies and reports [14,18,23]. Table 2 displays both the given mechanical strengths of timber and the corresponding estimated potential plastic coefficients.

#### 2.1.2. CFRP

The CFRP composite is represented using linear elastic isotropic behavior that fails when its ultimate tensile strength ($f_{u}$) is reached, taking into account the CFRP’s ability to withstand tensile stresses in the direction of fibers [7,24,25,26]. The ABAQUS software requires the definition of three essential parameters to properly describe the elastic model for the CFRP composite. The Poisson ratio is adopted at 0.3, while the modulus of elasticity $(E_{f}$) as well as the ultimate tensile strength ($f_{u}$) were considered according to experimental records available in the literature.

#### 2.1.3. Adhesive

The perfect bond model and the cohesive zone criterion are two independent techniques used in earlier numerical investigations to simulate the interfacial surface between the timber beam substrate and the CFRP layers. The study’s goal is to assess the two models’ accuracy and effectiveness in forecasting the load-deflection responses and capturing acceptable failure patterns while validating the FEM using experimental tests that are available in the literature. The most efficient technique will be implemented in the FEM used during the parametric study to show the effects of crucial factors on the flexural behavior of CFRP-strengthened timber beams.

##### Perfect Bond Model

To create a perfect bond model without allowing separation between the strengthening CFRP layers and the timber beam substrate, tie interaction is specified along the interfacial surface between the timber beam as the master surface and the CFRP layer as the slave surface.

##### Cohesive Zone Criterion

Unlike the perfect bond model, the cohesive zone criterion allows for movement on the adhesive surface between the timber beam substrate and the surface of the CFRP sheets, which can easily and efficiently predict the debonding of the CFRP sheets, if it occurs. A surface-to-surface contact interaction model is formed to represent the cohesive zone criterion between the timber beam substrate as the master surface and the CFRP sheets as the slave one. Several previous finite element models recommended the use of a bi-linear traction-separation behavior to simulate the cohesive surface between timber and CFRP thanks to its easy description, reduced computational time and cost, and advanced accuracy compared to exponential or linear exponential models [27,28,29,30,31,32].

The bi-linear traction-separation behavior described in Equation (14) requires the definition of three basic characteristics: prior-to-damage zone, damage-initiation point, and post-damage zone, as shown in Figure 1. The linear behavior of the prior-to-damage zone is expressed in terms of the initial stiffness $k_{nn}$ in the normal direction as well as $k_{ss}\;and\;k_{tt}$ for the shear directions using Equation (15) and Equation (16), respectively, according to the recommendations of Guo et al. [33] and Sakr [34].
(14)$$\left\{\begin{array}{c} \sigma_{n}\\ \tau_{s}\\ \tau_{t} \end{array}\right\}=\left[\left.\begin{array}{ccc} K_{nn} & 0 & 0\\ 0 & K_{ss} & 0\\ 0 & 0 & K_{tt} \end{array}\right]\right.\left\{\begin{array}{c} \delta_{n}\\ \delta_{s}\\ \delta_{t} \end{array}\right\}$$
(15)$$K_{nn}=\frac{E_{i}}{t_{i}}$$
(16)$$K_{ss}=K_{tt}=\frac{G_{i}}{t_{i}}=\frac{E_{i}}{2t_{i}(1+\upsilon_{i})}$$
where $\sigma_{n}$ represents the traction stress in the normal orientation; $\tau_{s}{\;and\;\tau }_{t}$ are the traction stress in the shear orientations; $\delta_{n},\;\delta_{s},{and\;\delta }_{t}$ stand for the associated displacement in the normal and shear directions, respectively; and $E_{i},\;{t_{i},G}_{i},and\;\upsilon_{i}$ are the elasticity modulus, thickness, shear modulus, and Poisson’s ratio of the adhesive, respectively.

The second phase in the cohesive zone criterion model requires the definition of the maximum normal strength along the interfacial surface$\;{(\sigma }_{nmax})$, which is specified as the minimal value between the tensile strength of the adhesive material and the yield stress of the timber material ($\sigma^{0})$. In addition, the input data in this phase should contain the identification of the maximum shear strength along the interfacial surface $\tau_{smax\;}{and\;\tau }_{tmax}$ in order to form the governing framework for the beginning of the damage in the adhesive surface at what is known as the damage-initiation point. When the calculated normal/shear stress reaches the corresponding maximum value, the damage is expected to start, as specified in the quadratic equation Equation (17). Furthermore, the analytical model provided by Lu et al. [35] was considered in the current FEM to calculate the maximum shear strength along the adhesive layer, as shown in Equation (18).
(17)$${\left\{\begin{array}{c} \frac{\sigma_{n}}{\sigma_{nmax}} \end{array}\right\}}^{2}+{\left\{\begin{array}{c} \frac{\tau_{s}}{\tau_{smax}} \end{array}\right\}}^{2}+{\left\{\begin{array}{c} \frac{\tau_{t}}{\tau_{tmax}} \end{array}\right\}}^{2}=1.0\;\;\;$$
(18)$$\tau_{smax}=\tau_{tmax}=1.12\sigma_{nmax}\;\;\;\leq 3.0\;MPa$$

The third characteristic of the cohesive model, the post-damage zone, was represented in this research using the fracture energy ($G_{f}$) criterion defined in Equation (19) along with the Benzeggagh–Kenane behavior [25]. For the reader’s information, the Benzeggagh–Kenane model can be applied when the fracture energies in the shear directions are equivalent ($G_{s}=G_{t})$ [14].
(19)$$G_{n}+{(G}_{s}-G_{n}){\left\{\frac{G_{S}}{G_{T}}\right\}}^{\eta }=G_{f}$$
where $G_{n}$ represents the fracture energy in the normal direction, $G_{S}=G_{s}+G_{t}$, $G_{T}=G_{n}+G_{s}$, and η is a material coefficient. Considering the conclusions from technical reports and experimental and numerical investigations [7,36,37,38,39,40,41], the fracture energies in both the normal and the shear directions are assumed to be within the range of 300–1500 J/m^2^, while η is set to be 1.45.

### 2.2. Elements, Mesh Size, and Boundary Conditions

In the current finite element model, the continuum three-dimensional eight-node linear brick element (C3D8) with six degrees of freedom (DOF) at every individual node is utilized to simulate the timber beam, the CFRP sheets, and the steel plates, as depicted in Figure 2. The main purpose of the steel plates is to distribute the pressure load over a larger area compared to the concentrated loads in order to avoid the concentration of stresses at loading points and around the boundary conditions (supports), which could lead to a local failure of the timber beam. The steel plates were represented using elastic response so they would not be deformed during loading. The assembled steel plates were joined to the timber beam and the CFRP sheets using a perfect bond model (tie constraint interaction).

All the elements of the timber beam, the CFRP layers, and the steel plates were discretized using a fine constant mesh size of 10 mm × 10 mm to facilitate the load transition between the assembled parts and to avoid any errors due to convergence problems during the running of the job. The boundary conditions of the current model contain two constraints. One of them is a hinge, and all of the displacements in the x, y, and z directions are restrained, as well as the rotation about the x direction (according to the model’s orientation). The second one is a roller support that prevents the displacement in the y and z directions, in addition to the rotation around the x direction, as shown in Figure 3. It should be mentioned that the job is terminated for all the proposed models (elastic, elastic-plastic, perfect bond, and cohesive zone criterion) at the maximum mid-span deflection experimentally recorded so that the failure load of timber beams strengthened with CFRP sheets can be logically compared.

### 2.3. Validation Using Available Experimental Studies

The proposed numerical model was evaluated in two steps. The first step is to analyze only timber beams without strengthening to evaluate the two proposed models for the representation of the timber material: the elastic and the elastic-plastic models. By the end of this step, the study aims to select the optimal model that successfully predicts the load-midspan deflection response, ultimate load capacity, as well as the failure pattern of bare timber beams compared to the experimental findings, which is suggested for representing the timber in the second phase.

The second phase consists of simply validating the FEM using timber beams externally strengthened with CFRP in order to evaluate the efficiency of two different models representing the adhesive layer between the timber beam substrate and CFRP sheets, namely the perfect bond model and the cohesive zone criterion. At the end of the first and second phases, the final optimum form of the numerical model to be used in the parametric study will be more evident through the conclusions of the different models nominated.

#### 2.3.1. Timber Beams

Three bare timber beams had different geometries and dimensions, as shown in Table 3. The parameters were available in previous studies [9,18,20], and they would be analyzed to verify the numerical model during the first phase.

Figure 4 compares the numerical load-midspan deflection responses for the analyzed timber beams using elastic and elastic-plastic models for the experimental data. The results revealed that the load-midspan deflection behavior of bare timber beams represented by the elastic model is always completely linear and the value of the applied load is continuously rising. In contrast, the experimentally tested timber beam behaviors are initially linear, followed by a non-linear region along with a decrease in the slope of the load-deflection curves. By comparing the behavior of the timber beams represented by the elastic model with the experimental records, it was found that they are compatible only in the linear phase and completely separate when the slope of the experimental curve is changed. This is due to the absence of a definition of the maximum limits of the stresses of timber material within this model, which led to the continuity of increasing the load with deflection, as shown in Figure 4.

The load-midspan deflection curves for bare timber beam specimens numerically produced using the elastic-plastic simulation of the timber material are completely consistent with the experimental findings, in contrast to the elastic model. The elastic stage, the nonlinear area, and the decrease in the slope of the load-deflection curves were all accurately anticipated by the elastic-plastic model. This is a result of the description of elastic and plastic characteristics, which also includes the definition of the yield stress for the timber material and is crucial to the discontinuous growth of the applied load. These results and ideas are compatible with the numerical and experimental failure patterns accompanied by the stress distribution along the length of the timber beams at the time of failure described in Figure 5, Figure 6 and Figure 7. The findings demonstrate that the stresses carried by the timber beams are higher when using the elastic model compared to the elastic-plastic model. For instance, the stress at failure for timber beam specimens A, A-NA-NA, and B0 represented using the elastic model was 227, 177, and 108 MPa, respectively, whereas their counterparts simulated by the elastic-perfect-plastic model achieved a stress level at the same maximum mid-span deflection of 62, 86, and 63 MPa, respectively.

The two models (elastic and elastic-plastic) were able to accurately predict the flexural failure of the timber beams as occurred in experiments. Specifically, the elastic-plastic model is more successful in concentrating the maximum stresses (flexural cracks) in the region of the maximum bending moment, as in the experimental tests, unlike the elastic model, which spreads the maximum stresses throughout the whole length of the beam specimens. Table 4 shows that the ratios of the ultimate loads obtained from the elastic model to the experimental values are in the range of 1.27 to 1.65, with an average value, standard deviation, and coefficient of variation of 1.49, 0.20, and 13.2%, respectively. Additionally, the ratios of the ultimate loads predicted by the elastic-plastic model to the experimental data are in the range of 0.99 to 1.05, with an average value, standard deviation, and coefficient of variation of 1.03, 0.03, and 3.1%, respectively. Moreover, the average ratio of both the elastic and the elastic-plastic maximum mid-span deflections to the experimental results is 1.03, with a standard deviation and coefficient of variation of 0.006 and 0.6%, respectively.

#### 2.3.2. Timber Beams Strengthened with FRP

Due to its high accuracy in predicting the flexural behavior of bare timber beams compared to the elastic model, the elastic-plastic model is used in the second phase, which is intended to ensure the accuracy of the numerical FEM by analyzing the structural response of timber beams strengthened with CFRP and GFRP sheets. Furthermore, the adhesive layer between the timber beam substrate and the FRP surface is represented by two different schemes (the perfect bond model and the cohesive zone criterion) in order to assess the accuracy of each model with respect to the experimental findings.

Five FRP-strengthened timber beams that had already undergone experimental testing and were accessible in earlier investigations were chosen to carry out the proposed numerical models’ validation process. It was necessary that the characteristics of these beams differ among themselves in terms of the fibers used, the number of strengthening layers, the location of the applied load, and the strengthening configurations, which could either be strengthening in the tensile region only using flat sheets or strengthening in the tensile and compression regions using U-shaped sheets, in order to increase the base of comparison so that the judgment on the accuracy of the proposed model can be logical. Table 5 demonstrates the characteristics of the FRP-strengthened timber beams as well as the data required to establish the cohesive zone model.

Figure 8 illustrates the entire compatibility between the experimental and numerical curves of load-midspan deflection of FRP-strengthened timber beams in the linear stage (elastic zone). In comparison to the results of the perfect bond model curves, in which the load-deflection response is linear for most of the modelled beams, the cohesive zone criterion model is more accurate in predicting the behavior of the analyzed specimens. With regard to the experimental findings, the cohesive model successfully captured both the linear and non-linear stages, particularly as the applied stress was increased and bending cracks began to form.

The effectiveness and accuracy of the used numerical model are substantially impacted by the pattern of collapse. The perfect bond model, in contrast to the cohesive zone model, failed to correctly estimate the ultimate loads of specimens F-U-U, B1, B2, and B-NA-1, as shown in Figure 9, Figure 10, Figure 11, and Figure 12, respectively. This is due to the fact that these FRP-enhanced timber beams collapsed as a result of FRP sheets debonding from timber substrates at the level of the adhesive layer or even with a thin layer of timber. The numerical axial stress analysis revealed that the four beams (F-U-U, B1, B2, and B-NA-1) represented by the perfect bond model ruptured because the stress level in the FRP sheets surpassed their ultimate tensile strength. Contrarily, in the case of these beams’ representation using the cohesive zone model, the interfacial shear stress value exceeded the maximum shear strength values previously defined, leading to the debonding of the strengthening sheets from the surface of the timber beams and accurately simulating the experimental results.

At first sight, it may appear that the two numerical models proposed to represent the behavior of FRP-strengthened timber beams accurately predicted the load-deflection curve of specimen B-NA-1 as depicted in Figure 8a. However, with the scrutiny of the curve, it is noted that the applied load in the case of the cohesive model began to abruptly drop at the maximum deflection of 39 mm because of the separation of the FRP sheets as shown in Figure 12. Conversely, the applied load at the same deflection value in the case of the perfect bond model continues to increase. The convergence of the ultimate load value of the two numerical models came as a result of the termination of the numerical job at the maximum mid-span deflection that was experimentally recorded.

The perfect bond model was able to predict the flexural cracks and the rupture that occurred in the timber beams as well as the cohesive zone model, even though it was unable to predict the pattern of the debonding of the strengthening FRP sheets from the timber beams and the value of the predicted ultimate load compared to the experimental records was exaggerated. Therefore, the perfect bond model was able to predict with high efficiency the load-deflection response and the ultimate load of the D-NA-U specimen, which collapsed as a result of the cracking of the timber beams in the compression zone, as shown in Figure 13.

According to the statistical analysis shown in Table 6, which is consistent with the above discussions, the results of the cohesive zone model are more in line with the experimental data than the results of the perfect bond model. The ratios between the perfect bond model ultimate loads and the experimental records are in the range of 1.04 to 1.90, with an average value, standard deviation, and coefficient of variation of 1.38, 0.43, and 31.3%, respectively. Additionally, the ratios between the ultimate loads predicted by the cohesive zone model and the experimental data are in the range of 0.97 to 1.04 with an average value, standard deviation, and coefficient of variation of 1.00, 0.025, and 2.54%, respectively. Moreover, the average ratio between both the perfect bond and the cohesive zone maximum mid-span deflections and the experimental results is 1.02, with a standard deviation and coefficient of variation of 0.034 and 3.28%, respectively.

## 3. Parametric Analysis

The numerical model, which has been validated for its accuracy and ability to predict the load-deflection behavior, failure modes, and maximum load capacity of timber beams strengthened in flexure with CFRP sheets, was used to conduct a parametric study that benefits design engineers and those interested in the behavior of such structures. The main objective of the parametric analysis is to investigate how the length, position, and thickness of the strengthening sheets affect the flexural characteristics of the timber beams strengthened by the CFRP sheets. Twelve samples were used in which the ratio between the length of the CFRP strengthening sheets and the total length of the timber beam ranged from 50% to 100%, with an increment rate of 10%. Additionally, the location of the strengthening sheets was changed, either in the tensile zone only or in both the tensile and compressive zones (bottom and upper sides, respectively).

It is worth noting that the thickness of the strengthening sheet is constant for the twelve specimens and is equal to 1.0 mm to represent the strengthening with only one layer, whether on the bottom side only or on the bottom and upper sides together. Moreover, the behavior of a timber beam (B2-L-T) strengthened with two layers of CFRP sheets with a total thickness of 2.0 mm at the bottom side only was analyzed. All of the aforementioned timber beams’ responses were compared to the behavior of the un-strengthened specimen (B-0-0).

The timber beams that Nadir et al. [9] experimentally tested and strengthened using CFRP sheets, which have a higher tensile strength than the GFRP sheets, were the basis for the parametric analysis. Table 7 presents the numerical program for the analyzed specimens, indicating the length, location, and thickness of the strengthening CFRP sheets, while Figure 14 shows the strengthening configurations.

The effects of different lengths of CFRP sheets to strengthen timber beams on failure patterns, ultimate loads, flexural rigidity, and ductility index are shown in Table 8 and addressed in the research’s subsequent sections.

### 3.1. Description of the Failure Patterns

For the bare beam (B-0-0), flexural failure occurred at ultimate load due to the rupture of the timber beam on the tensile fiber, as previously depicted in Figure 5, Figure 6 and Figure 7. On the other hand, all the timber beams strengthened by CFRP sheets collapsed as a result of the debonding of the strengthening layers from the timber substrates, even as the length and place of the strengthening layers varied, depending on whether they were only on the tensile side or on both the tensile and compressive sides, as shown in Figure 15a and Figure 15b, respectively. Although the timber beams that were strengthened with CFRP sheets in both the tensile and compressive zones collapsed as a result of the debonding of the strengthening sheets from the timber substrates, the separation occurred at higher ultimate loads with respect to the specimens strengthened in the tensile zone only.

The main factor that led to the delay in the debonding of the strengthening CFRP sheets from the surfaces of the timber beams and the increase in the ultimate load is the increase in the cohesion area on the surface between the strengthening sheets and the timber beams when using CFRP layers in the tensile and compressive zones together rather than using one CFRP layer in the tensile zone only. In addition, the increase in thickness of the strengthening sheets in the case of two strengthening sheets in both the tensile and compressive zones, which was twice the thickness of the strengthening sheets used in the tensile position only, increased the strengthening area, which in turn reduced the applied shear stress on the interfacial surface between CFRP and timber. Moreover, the flexural rigidity of timber beams strengthened with CFRP in both the tensile and compressive zones was significantly higher than that of those strengthened in the tensile zone only.

### 3.2. Load-Deflection Responses

Figure 16 displays the findings of the load-midspan deflection correlation curves for timber beams strengthened with various CFRP sheet lengths. For all timber beams strengthened with CFRP sheets, either in the tensile zone only or in both the tensile and compressive zones, as the length of the CFRP sheet increased, the gain in ultimate load increased compared to the un-strengthened timber specimen. This is due to the easy transmission of load between loading points and supports, especially as the length of the strengthening CFRP sheets increases. This effectively delays the separation of the CFRP sheets by transferring the ends of their connection to timber beams outside the maximum moment limits. Although the use of longer CFRP sheets prevented their debonding as a result of the initiation and widening of cracks in the maximum moment zone, they were separated as a result of high shearing stresses affecting the interfacial surface between the strengthening sheets and timber beams located near the supports.

The findings strongly support the main objective of this research. In comparison to the load of timber beams CFRP-strengthened in the tensile zone only, the ultimate load of the beams that were strengthened in both the tensile and compressive zones was significantly higher. The gain in ultimate load of timber beams strengthened with CFRP sheets in the tensile zone only ranged from 10.3% to 30.9%. These percentages did not fluctuate consistently with the change in the length of the strengthening sheets. The value of the ultimate load increased as the length of the strengthening CFRP sheets was extended up to 80% of the length of the timber beams; however, from 80% to 100% of the length of the timber beams, there was no discernible increase in the value of the ultimate load. Therefore, it is possible to consider the effective strengthening length as being 80% of the length of the timber beams that have only been strengthened in the tensile zone with CFRP sheets.

The ultimate load of the timber beams strengthened with CFRP sheets in the tensile and compressive regions together is directly proportional to the increase in the lengths of the strengthening sheets. The increase in the maximum load for the specimens strengthened by CFRP sheets that were 50% to 100% of the lengths of the timber beams ranged from 27.9% to 53.7%. As evidenced by the slope of the curves presented in Figure 16, the strengthening CFRP sheets installed in both the tensile and compressive zones were successful in significantly increasing the value of the flexural rigidity of the timber beams. This increased the value of the total flexural capacity of these beams compared to the timber beams strengthened in the tensile zone only. It is worth noting that there was no significant difference in the value of the ultimate load between the sample strengthened by two layers of CFRP sheets in the tensile zone (B2-L-T) and the sample strengthened by one layer in the tensile zone together with another layer in the compressive zone (B-L-T&C). All the CFRP-strengthened specimens showed deflection values at failure greater than the un-strengthened timber beam (B-0-0), except for the sample that was strengthened in the tensile region only with a sheet of CFRP equal to half the length of the timber beam (B-0.5L-T).

### 3.3. Flexural Rigidity

The flexural rigidity (EI) of the bare timber beam as well as the FRP-strengthened timber beams with different strengthening lengths was calculated at both the elastic and ultimate stages based on the experimentally recorded load-midspan deflection data according to Equation (20).
(20)$$EI=\frac{23PL^{3}}{1296\Delta }$$
where EI is the flexural rigidity (N · mm^2^); E is the modulus of elasticity (N/mm^2^); I is the moment of inertia (mm^4^); P and Δ are the applied load at the required stage (N) and the corresponding mid-span deflection (mm), respectively; and L is the effective span length (mm).

The findings in Table 8 demonstrate the effects of CFRP sheets to strengthen timber beams on the flexural rigidity of such structures, regardless of whether the strengthening layers are placed solely in the tensile zone or in both the tensile and compressive zones. Both techniques employed to strengthen the bare timber beams significantly enhanced the flexural rigidity at the elastic stage when compared to the un-strengthened specimen; however, the application of the external CFRP sheets in both the tensile and compressive zones demonstrated the highest values. This result, although previously known and predictable using preliminary structural analysis principles, explains the increased inclination of load-deflection curves as well as the ultimate load in CFRP-strengthened timber beams in both the tensile and compressive zones when compared to the un-strengthened specimen or the beams that were CFRP-strengthened in only the tensile zone. During the elastic stage, the flexural rigidity values of timber beams strengthened in the tensile zone only ranged from 11.60 × 10^9^ N · mm^2^ to 13.00 × 10^9^ N · mm^2^, whereas the same values for their counterparts strengthened in the tensile and compressive zones ranged from 16.28 × 10^9^ N mm^2^ to 20.7 × 10^9^ N · mm^2^, as opposed to 7.81 × 10^9^ N · mm^2^ for the un-strengthened control specimen. The gain in flexural rigidity of timber beams strengthened only in the tensile zone with different lengths and their counterparts strengthened in both the tensile and compressive zones with the same strengthening lengths was in the range of 49–66% and 108–165%, respectively, with respect to the control timber beam.

In addition to the values of flexural stiffness at the elastic stage, the flexural stiffness values were calculated again at the ultimate stage, specifically at the ultimate load, due to the deterioration of the condition of the timber beams between those two stages (linear and ultimate). As a result of the emergence and widening of flexural cracks in addition to the debonding of the CFRP sheets, the value of the flexural stiffness of the timber beams at the ultimate stage decreased compared to the linear one, which directly affects the value of the mid-span deflection at failure. This also explains why the applied load abruptly decreased in value after reaching its ultimate value, as indicated by the curves of the load-midspan deflection, as seen in Figure 16. The un-strengthened timber specimen collapsed early, and the ratio between the ultimate stage flexural stiffness and the elastic stage stiffness was 0.71. Strengthening the timber beams with CFRP sheets delayed the failure due to the resilience of strengthened beams and improved flexural stiffness values at the ultimate stage compared to the un-strengthened specimen. In timber beams strengthened in the tensile zone only using different lengths of CFRP sheets, the ratio between flexural stiffness in the ultimate and elastic stages was in the range of 0.46–0.55, compared to 0.31–0.4 for the specimens strengthened in both the tensile and compressive zones.

### 3.4. Ductility Index

Instead of only improving the ultimate load before collapse, it is necessary to evaluate the efficiency of CFRP sheets as a strengthening material by discussing the ductility results of the specimens modelled. In this research, the ductility index was determined as the ratio between the displacement at ultimate load (*u_u_*) and displacement at yield load (*u_y_*) based on the recommendations of the Swiss code for timber structures [42] as shown in Equation (21) and Figure 17.
(21)$$Ductilityindex=\frac{u_{u}}{u_{y}}$$

Table 8 displays the ductility results of the timber beams strengthened with CFRP sheets along with the increasing ratios over the reference beam that was not strengthened. In general, whether the location of the strengthening CFRP sheets was in the tensile zone only or in the tensile and compressive zones, the ductility of the timber beams improved with respect to the reference timber beam. Particularly, compared to timber beams that were strengthened in the tensile region alone, the results of the specimens strengthened in both the tensile and compressive regions showed a much better ductility before failure. Moreover, the results show that as the length of the strengthening CFRP sheets increases in the tensile region only or in both the tensile and compressive regions, the ductility increases as a result of improving the mid-span deflection values at ultimate load. On one hand, the increasing ratio in the ductility index of the timber beams strengthened by CFRP sheets in the tensile zone only ranged from 44.8%, when the strengthening sheet length was only equal to half of the timber beam length, to 126.9%, in the case where the length of the strengthening sheet was equal to the timber beam length, compared to the reference beam. On the other hand, specimens that had their tensile and compressive zones strengthened with CFRP sheets of various lengths, spanning from half to the full length of the timber beam, recorded an increase in ductility values of between 136.6 and 240.7% with respect to the reference specimen.

## 4. Conclusions

Extensive numerical investigations on the flexural behavior of timber beams strengthened with FRP under bending loads are conducted in this paper. Three-dimensional finite element models of timber beams externally strengthened with FRP are developed. At first, the accuracy of elastic and elastic-plastic models in simulating the flexural behavior of timber material is compared. Next, the efficiency of two different models in representing the interfacial behavior between the timber beam substrate and FRP sheets (i.e., the perfect bond model and the cohesive zone model) is evaluated. At last, the most accurate model in predicting the behavior of timber beams strengthened with FRP is used for a parametric study to assess the effects of length, thickness, and position of FRP on the failure mode, load-bearing capacity (i.e., flexural strength), flexural rigidity, and ductility index of the hybrid FRP–timber beams. Based on the results and discussions reported above, the following conclusions can be drawn:

Elastic-plastic models with the anisotropic Hill yield criterion correctly predict the failure pattern, flexural strength, and load-deflection behavior of bare timber beams, while elastic models overestimate the flexural strength of the beams.The perfect bond model exhibits an accurate prediction of the behavior of the strengthened beams when the failure of the beam was governed by the cracking in the timber material.All strengthened beams fail as a result of FRP debonding from the timber substrate regardless of the variation in the strengthening configuration and dimensions.The FRP length and thickness have a noticeable influence on the flexural strength of the strengthened beams, that is, the flexural strength of the strengthened beams increases as the span length and thickness of the FRP increase.The increases in the flexural strength of the beams range from 10.3% to 30.9% when the span length of the CFRP placed in the tensile zone varied from 50% to 100% of the total length of the beams.Utilization of a single layer and a double layer of CFRP sheets increases the flexural strength by 30.9% and 52.9%, respectively, compared to the un-strengthened beam.The increase in the flexural rigidity of timber beams was strengthened only in the tensile zone with increasing lengths, and their counterparts were strengthened in both the tensile and compressive zones with the same strengthening lengths, in the ranges of 49–66% and 108–165%, respectively, with respect to the control timber beam.Strengthening the beam in both the tensile and compressive zones exhibits an increase in the ductility index of 136.6% and 240.7% when the CFRP length is 50% and 100% of the total beam length, respectively.

In light of the above conclusions, for timber beam strengthened with FRP, it is recommended to (a) model the timber as an elastic-plastic material rather than elastic material and the FRP–timber interface behavior using the cohesive zone model to ensure a more realistic distribution of stress in the timber beam and accurate prediction of failure mode and flexural strength; (b) limit the FRP length to 80% of the total length of the beam; and (c) strengthen both the tensile and compressive zones of the beam when possible.

## Figures and Tables

**Figure 1 materials-17-00321-f001:**
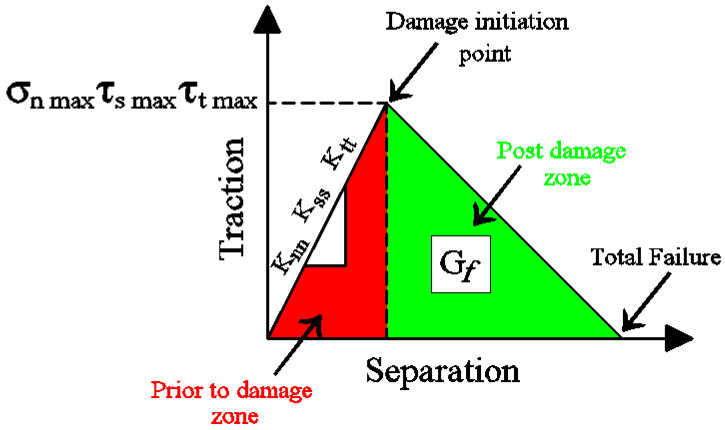
Bilinear traction-separation model used to define the cohesive zone criterion.

**Figure 2 materials-17-00321-f002:**
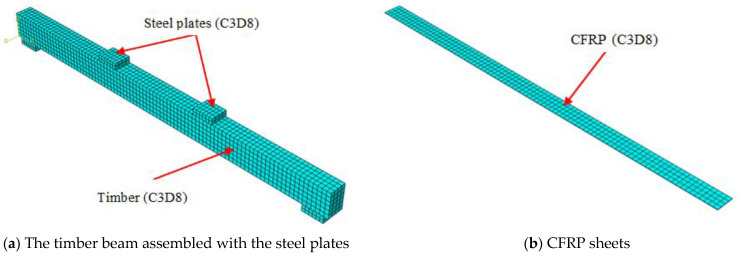
Elements and mesh pattern of the FEM.

**Figure 3 materials-17-00321-f003:**
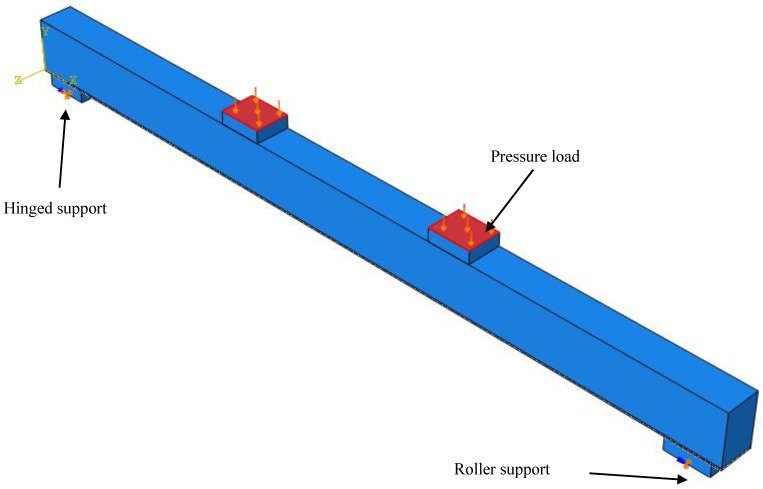
Loading and constraints of the proposed FEM.

**Figure 4 materials-17-00321-f004:**
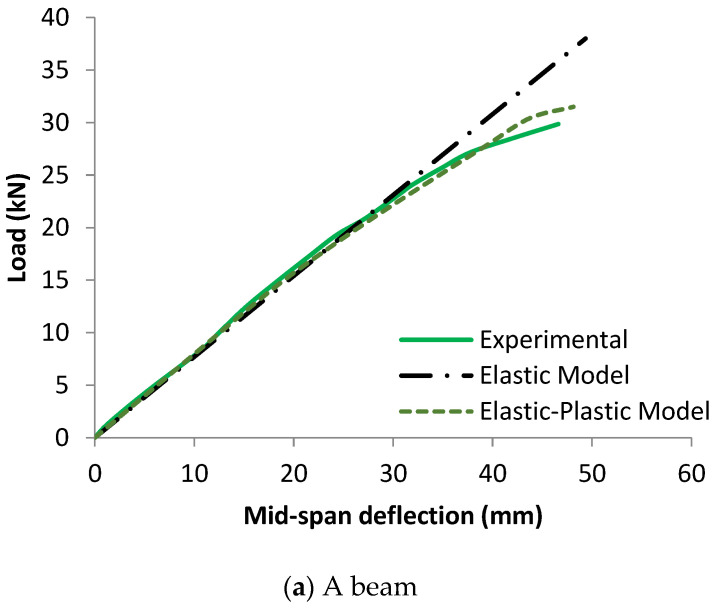
Effect of elastic and elastic-plastic models on the load-midspan deflection responses of timber beams.

**Figure 5 materials-17-00321-f005:**
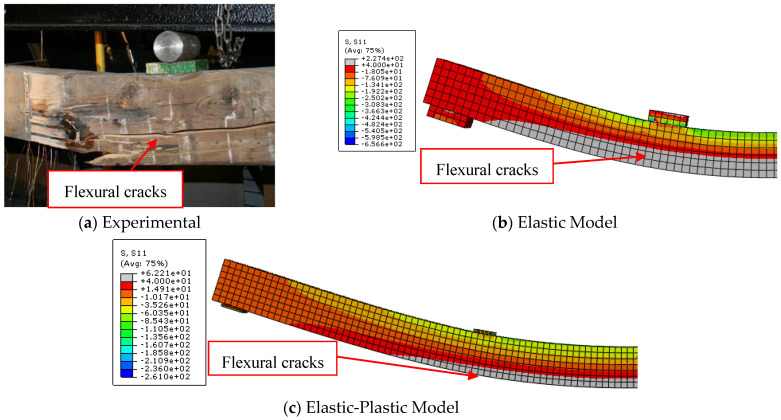
Comparison between experimental and numerical failure patterns of A specimen.

**Figure 6 materials-17-00321-f006:**
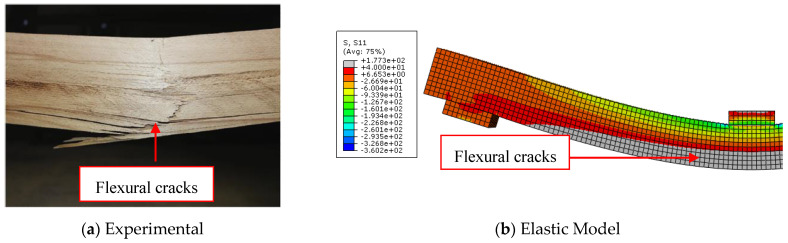
Comparison between experimental and numerical failure patterns of A-NA-NA specimen.

**Figure 7 materials-17-00321-f007:**
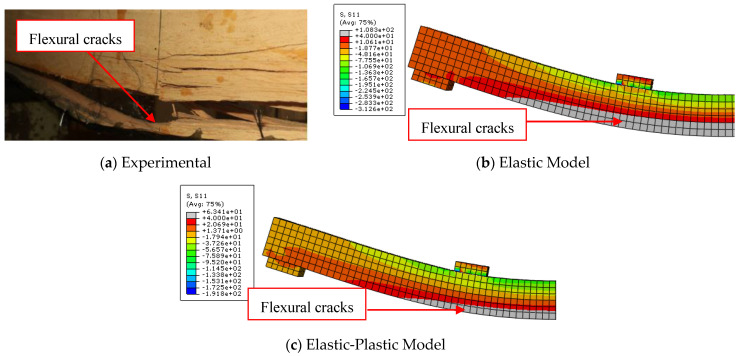
Comparison between experimental and numerical failure patterns of B0 specimen.

**Figure 8 materials-17-00321-f008:**
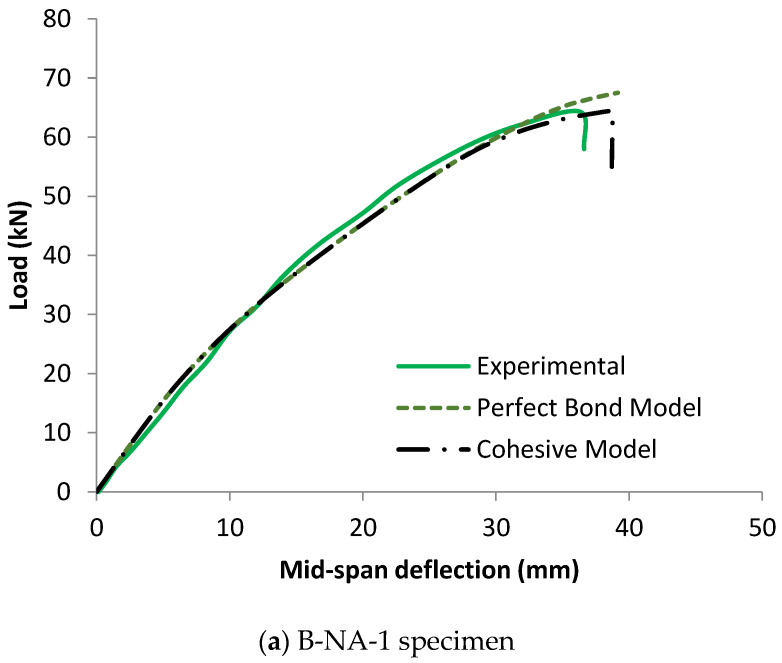
Effect of perfect bond and cohesive models on the load-midspan deflection responses of CFRP-strengthened timber beams.

**Figure 9 materials-17-00321-f009:**
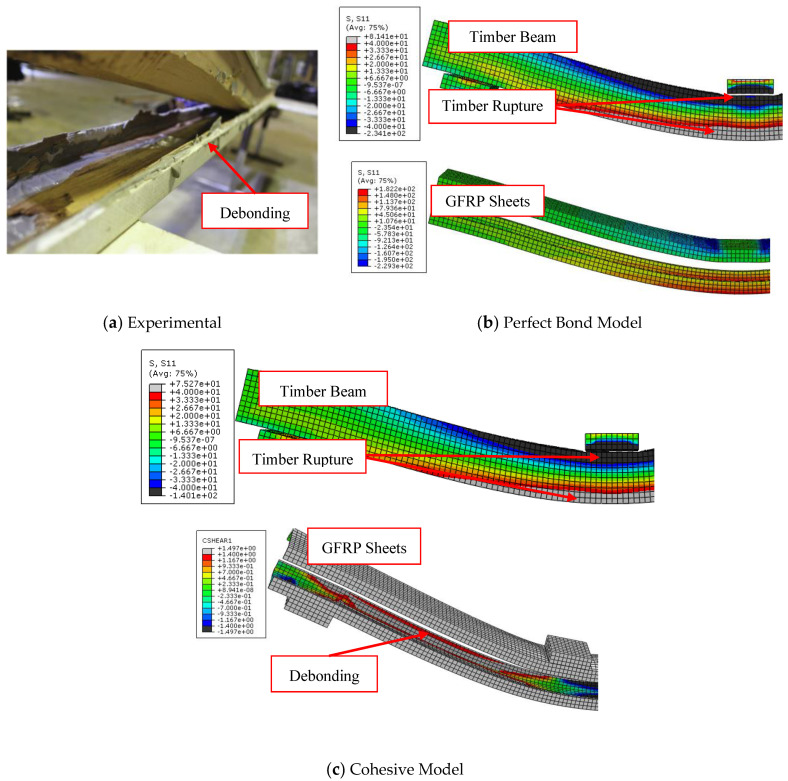
Comparison between experimental and numerical failure patterns of F-U-U specimen.

**Figure 10 materials-17-00321-f010:**
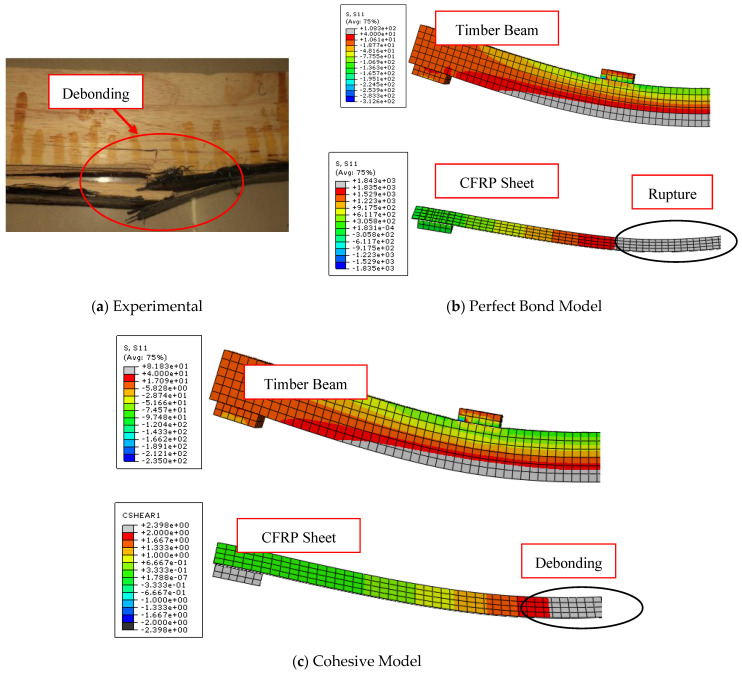
Comparison between experimental and numerical failure patterns of B1 specimen.

**Figure 11 materials-17-00321-f011:**
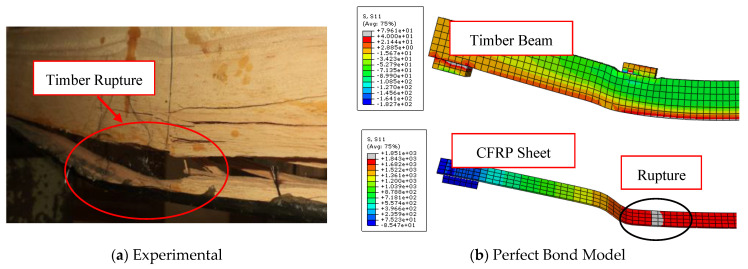
Comparison between experimental and numerical failure patterns of B2 specimen.

**Figure 12 materials-17-00321-f012:**
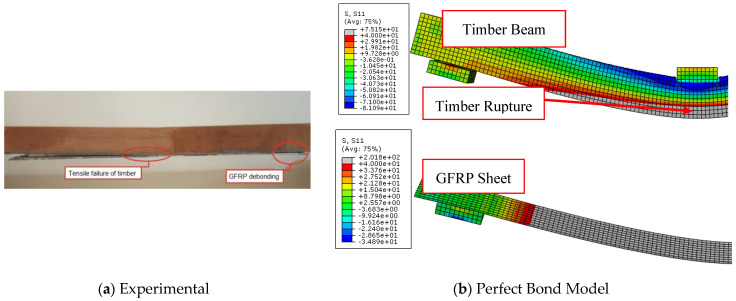
Comparison between experimental and numerical failure patterns of B-NA-1 specimen.

**Figure 13 materials-17-00321-f013:**
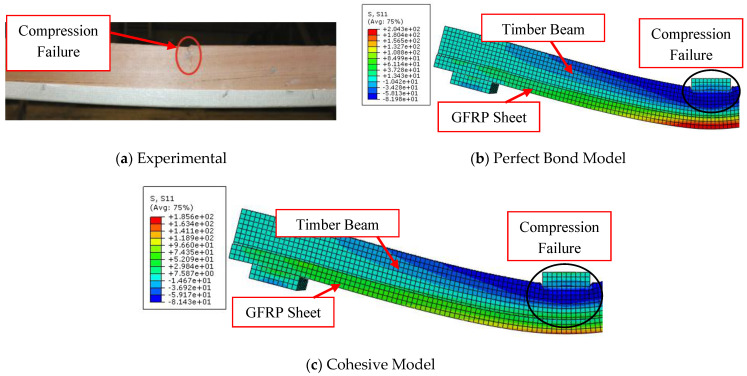
Comparison between experimental and numerical failure patterns of D-NA-U specimen.

**Figure 14 materials-17-00321-f014:**
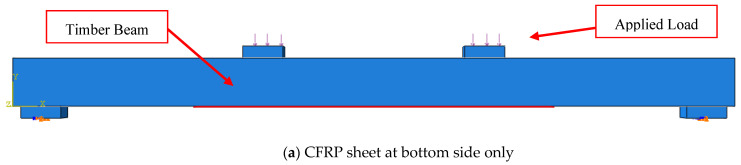
Strengthening configuration implemented in the parametric study.

**Figure 15 materials-17-00321-f015:**
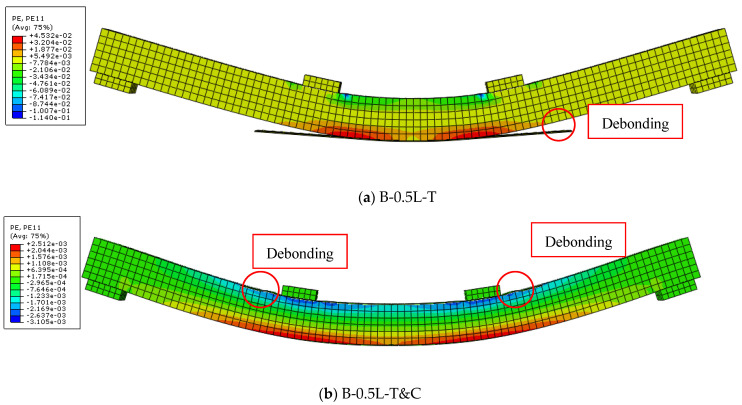
Debonding of the CFRP sheets from the timber substrates.

**Figure 16 materials-17-00321-f016:**
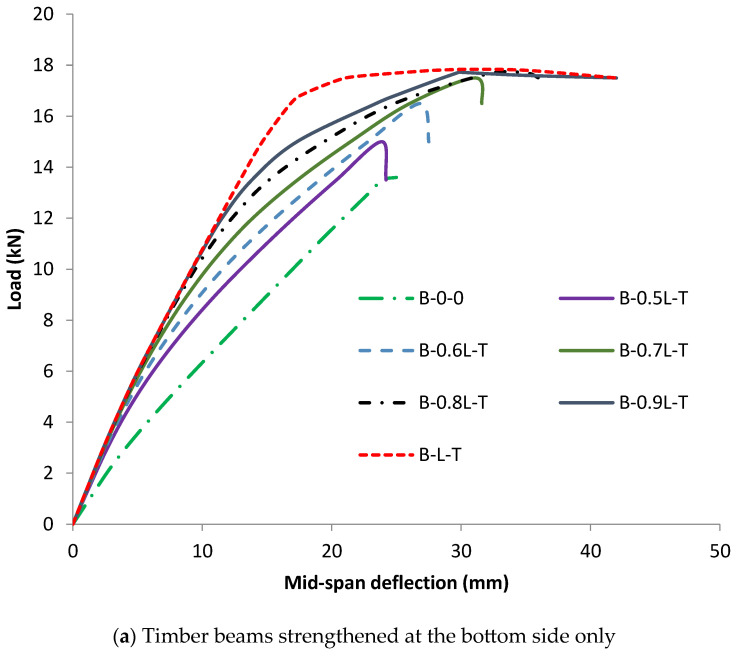
Effect of the strengthening CFRP sheet length and its position on the load-deflection response.

**Figure 17 materials-17-00321-f017:**
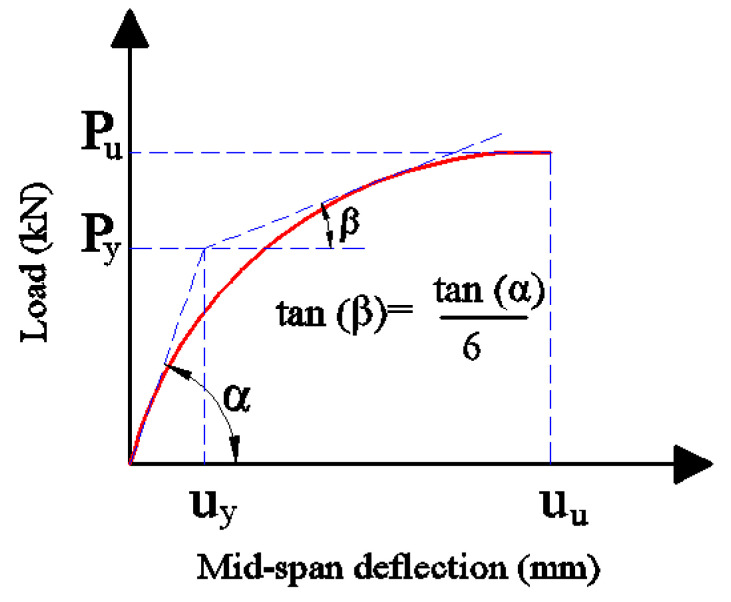
The method used to determine the ductility index values for the timber beams strengthened with CFRP sheets.

**Table 1 materials-17-00321-t001:** Engineering constants required to establish the elasticity model for timber.

$E_{1}(MPa)$	$E_{2}\;(MPa)$	$E_{3}\;(MPa)$	$\upsilon_{12}$	$\upsilon_{13}$	$\upsilon_{23}$	$G_{12}\;(MPa)$	$G_{13}\;(MPa)$	$G_{23}\;(MPa)$
11,439	320.4	320.4	0.013	0.013	0.23	600.75	600.75	600.75

**Table 2 materials-17-00321-t002:** Values of the mechanical strengths of timber and the calculated potential plastic coefficients.

$\sigma^{0}=\sigma_{11}$ (MPa)	$\sigma_{22}\;(MPa)$	$\sigma_{33}\;(MPa)$	$\sigma_{12}\;(MPa)$	$\sigma_{13}\;(MPa)$	$\sigma_{23}\;(MPa)$	$R_{11}$	$R_{22}=R_{33}$	$R_{12}=R_{13}$	$R_{23}$
40	10	10	15.5	15.5	7.6	1.0	0.25	0.67	0.33

**Table 3 materials-17-00321-t003:** Dimensions of the analyzed timber beams.

Reference	Beam ID	Width × Depth × Length mm	Shear Span mm	Shear Span/Depth Ratio
Nowak et al. [18]	A	120 × 220 × 4000	1270	5.7
Shekarchi et al. [20]	A-NA-NA	85 × 100 × 1400	600	6.0
Nadir et al. [9]	B0	40 × 60 × 900	276	4.6

**Table 4 materials-17-00321-t004:** Comparison between experimental findings (Exp) and numerical records (Num) for timber beams.

Beam ID			Ultimate Load (kN)			Maximum Mid-Span Deflection (mm)				
	Exp	Num (M1)	Num (M2)	M1/Exp	M2/Exp	Exp	Num (M1)	Num (M2)	M1/Exp	M2/Exp
A	29.9	38	31.5	1.27	1.05	46.6	48.1	48.1	1.03	1.03
A-NA-NA	56	92.2	55.4	1.65	0.99	32.5	33.5	33.5	1.03	1.03
B0	13	20	13.6	1.54	1.04	24.1	25	25	1.04	1.04
Average				1.49	1.03				1.03	1.03
Standard deviation				0.20	0.03				0.006	0.006
Coefficient of variation (COV)%				13.2	3.1				0.6	0.6

Notes: M1 represents elastic model; M2 represents elastic-plastic model.

**Table 5 materials-17-00321-t005:** Properties and defined data for the numerically analyzed FRP-strengthened timber beams using cohesive model.

Reference	Beam ID	Strengthening Pattern/ Material Type	FRP Thickness (mm)	Strengthening Position	Tensile Strength of FRP MPa	$K_{nn}$ N/mm^3^	$K_{ss}$ and $K_{tt}$ N/mm^3^	$\tau_{smax}$ and $\tau_{tmax}$ MPa	$G_{n}$ J/m^2^	$G_{s}$ and $G_{t}$ J/m^2^
Shekarchi et al. [20]	B-NA-1	Flat/GFRP	2.5	Bottom side	450	1300	500	1.5	900	400
	D-NA-U	U-shaped/GFRP	2.5	Bottom side	450	1300	500	1.5	900	400
	F-U-U	U-shaped/GFRP	2.5	Bottom and upper sides	450	1300	500	1.5	900	400
Nadir et al. [9]	B1	Flat/CFRP	1.0	Bottom side	1835	2600	1000	2.4	1300	500
	B2	Flat/CFRP	2.0	Bottom side	1835	2600	1000	2.4	1300	500

**Table 6 materials-17-00321-t006:** Experimental (Exp) versus numerical (Num) ultimate load and maximum deflection for timber beams strengthened with FRP.

Beam ID	Ultimate Load (kN)					Maximum Mid-Span Deflection (mm)				
	Exp	Num (M3)	Num (M4)	M3/Exp	M4/Exp	Exp	Num (M3)	Num (M4)	M3/Exp	M4/Exp
B-NA-1	64.3	67.5	64.5	1.05	1.00	36.6	39.2	39.2	1.07	1.07
D-NA-U	73	76.3	71	1.04	0.97	42.1	44	44	1.04	1.04
F-U-U	92	102	92	1.11	1.00	37	36.4	36.4	0.98	0.98
B1	18	33	17.8	1.80	0.99	41.3	42	42	1.02	1.02
B2	20	38.3	20.8	1.90	1.04	41.4	42	42	1.01	1.01
Average				1.38	1.00				1.02	1.02
Standard deviation				0.43	0.025				0.034	0.034
Coefficient of variation (COV)%				31.3	2.54				3.28	3.28

Notes: M3 represents perfect bond model; M4 represents cohesive model.

**Table 7 materials-17-00321-t007:** The total program of the numerically analyzed beams in the parametric study.

Beam ID	Strengthening Pattern/ Material Type	FRP Thickness (mm)	Strengthening Position	CFRP Sheet Length in Bottom and Upper Sides (mm)	CFRP Sheet Length/Timber Beam Length
B-0-0	-	-	-	-	-
B-0.5L-T	Flat/CFRP	1.0	Bottom side	450	0.5
B-0.5L-T&C	Flat/CFRP	1.0	Bottom and upper sides	450	0.5
B-0.6L-T	Flat/CFRP	1.0	Bottom side	540	0.6
B-0.6L-T&C	Flat/CFRP	1.0	Bottom and upper sides	540	0.6
B-0.7L-T	Flat/CFRP	1.0	Bottom side	630	0.7
B-0.7L-T&C	Flat/CFRP	1.0	Bottom and upper sides	630	0.7
B-0.8L-T	Flat/CFRP	1.0	Bottom side	720	0.8
B-0.8L-T&C	Flat/CFRP	1.0	Bottom and upper sides	720	0.8
B-0.9L-T	Flat/CFRP	1.0	Bottom side	810	0.9
B-0.9L-T&C	Flat/CFRP	1.0	Bottom and upper sides	810	0.9
B-L-T	Flat/CFRP	1.0	Bottom side	900	1.0
B-L-T&C	Flat/CFRP	1.0	Bottom and upper sides	900	1.0
B2-L-T	Flat/CFRP	2.0	Bottom side	900	1.0

**Table 8 materials-17-00321-t008:** Effect of the strengthening length on the ultimate load, maximum deflection, flexural rigidity, ductility index, and failure mode of the CFRP-strengthened timber beams.

Beam ID	Ultimate Load (kN)	Increasing Ratio %	Maximum Mid-Span Deflection (mm)	Flexural Rigidity (N · mm^2^)			Ductility Index	Increasing Ratio %	Failure Pattern
				Elastic Stage	Ultimate Stage	Ultimate /Elastic			
B-0-0	13.6	-	25.0	7.81 × 10^9^	5.52 × 10^9^	0.71	1.45	-	Flexural failure
B-0.5L-T	15	10.3	24.2	11.60 × 10^9^	6.38 × 10^9^	0.55	2.10	44.8	Debonding
B-0.5L-T&C	17.4	27.9	27.5	16.28 × 10^9^	6.43 × 10^9^	0.40	3.43	136.6	Debonding
B-0.6L-T	16.5	21.3	27.5	12.25 × 10^9^	6.23 × 10^9^	0.51	2.13	46.9	Debonding
B-0.6L-T&C	18.5	36.0	29.7	18.30 × 10^9^	6.38 × 10^9^	0.35	4.37	201.4	Debonding
B-0.7L-T	17.5	28.7	31.6	12.67 × 10^9^	6.10 × 10^9^	0.48	2.85	96.6	Debonding
B-0.7L-T&C	18.7	37.5	28.1	19.74 × 10^9^	6.63 × 10^9^	0.34	4.69	223.4	Debonding
B-0.8L-T	17.7	30.1	36.0	12.91 × 10^9^	6.08 × 10^9^	0.47	2.97	104.8	Debonding
B-0.8L-T&C	19.1	40.4	30.7	20.46 × 10^9^	6.26 × 10^9^	0.31	4.72	225.5	Debonding
B-0.9L-T	17.7	30.1	42.0	12.99 × 10^9^	6.02 × 10^9^	0.46	3.27	125.5	Debonding
B-0.9L-T&C	19.3	41.9	29.3	20.67 × 10^9^	6.48 × 10^9^	0.31	4.91	238.6	Debonding
B-L-T	17.8	30.9	42.0	13.00 × 10^9^	5.98 × 10^9^	0.46	3.29	126.9	Debonding
B-L-T&C	20.9	53.7	32.0	20.7 × 10^9^	6.62 × 10^9^	0.32	4.94	240.7	Debonding
B2-L-T	20.8	52.9	42.0	16.10 × 10^9^	6.52 × 10^9^	0.41	4.65	220.7	Debonding

## Data Availability

The raw data required to reproduce these findings are available only with direct contact through email to the corresponding author. The processed data required to reproduce these findings are available only with direct contact through email to the corresponding author.

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
