# Peer review of "Improving the Flexural Response of Timber Beams Using Externally Bonded Carbon Fiber-Reinforced Polymer (CFRP) Sheets"

_materials, 2024, doi:10.3390/ma17020321_

Round 1

Reviewer 1 Report

Comments and Suggestions for Authors

The manuscript is well written. All parameters are thoroughly explained and the methodology is clear. The numerical model is firstly validated against available data from the scientific literature before the parametric study was conducted.

Please address the following issues in order to make your work clearer to the readers.

Line 98 - the authors refer to 3-point bending test (reference 13) which is later on used for the validation of their numerical model (Table 5). However, the numerical model employs 4-point loading / bending test. Please make the necessary correction.

Line 125 - the paper presents only numerical analyses on 4-point loading tests.

Line 183 - there is an extra "=" sign

Line 190 - equations 10-13

To my understanding, the authors used the values of the mechanical properties (strength and elastic) from scientific literature. However, the values presented in Table 1 use different sources from the values presented in Table 2 (except reference 11). Although they refer to two distinct numerical models of timber, the different sources may lead to results that could not be directly compared to one another.

Line 268 - there are two notations of the fracture energy that are not explained in the text.

Please double check the figures to make sure all text is visible.

Please reconsider the Conclusions section and make it more concise. Additionally, it is advisable to use present tens throughout this section.

Comments on the Quality of English Language

Lines 161,162 - please substitute "modulus" by "moduli" 

Line 208 - "were considered"

Line 501 - please substitute "erupted" by "occurred"

Line 512 - please substitute "to" by "the"

Author Response

Manuscript ID: materials-2810909

Improving the flexural response of timber beams using externally bonded carbon fibre-reinforced polymer (CFRP) sheets

Dear Editor,

The authors appreciate the suggestions of the reviewer aiming at enhancing the quality of the paper. The valuable comments have been considered in the revised manuscript as elaborated below:

Reviewer (1):

  • Line 98 - the authors refer to 3-point bending test (reference 13) which is later on used for the validation of their numerical model (Table 5). However, the numerical model employs 4-point loading / bending test. Please make the necessary correction.
  •  The reference No. 13 used 3-point bending test not 4-point bending test.
  • Line 125 - the paper presents only numerical analyses on 4-point loading tests.
  • Edited as required.
  • Line 183 - there is an extra "=" sign
  • It is not found in the submitted manuscript. It is a typos generated by the submission system.
  • Line 190 - equations 10-13: To my understanding, the authors used the values of the mechanical properties (strength and elastic) from scientific literature. However, the values presented in Table 1 use different sources from the values presented in Table 2 (except reference 11). Although they refer to two distinct numerical models of timber, the different sources may lead to results that could not be directly compared to one another.
  • Table 1 represents the elastic parameters of wood beams while Table 2 represents the plastic properties and there is no comparison between the two tables because each has a different function. Also, the values considered from previous research do not conflict with the experimentally recorded characteristics defined in Table 2 because Table 2 defines variables for which values are not defined in the mentioned equations (10-13).
  • Line 268 - there are two notations of the fracture energy that are not explained in the text.
  • All fracture energies were defined through lines from 267 to 270.
  • Please double check the figures to make sure all text is visible.
  • All figures has been checked as required, all text is visible.
  • Please reconsider the Conclusions section and make it more concise. Additionally, it is advisable to use present tens throughout this section.
  • Conclusions section has been totally re-written as required.
  • Lines 161,162 - please substitute "modulus" by "moduli"
  • Edited as required.
  • Line 208 - "were considered"
  • Edited as required.
  • Line 501 - please substitute "erupted" by "occurred"
  • Edited as required.
  • Line 512 - please substitute "to" by "the"
  • Edited as required.

Reviewer 2 Report

Comments and Suggestions for Authors

The manuscript presents numerical (FE) analysis on FRP reinforced timber beams including validation based on literature data and a parametric analysis, focusing on wood mechanical modeling and bond modeling.

The authors should address some minor issues and make some improvements as detailed below.  

The title is too general. A more specific title to fit the contents of the work might be considered.  

The literature survey has to be extended with a few relevant items of recent years as follows. A review article on reinforcing timber with FRP is given by https://doi.org/10.3390/polym14122381, which should be cited. Another review is given by http://dx.doi.org/10.32604/jrm.2022.021983. The authors address bonding in line 59. The first mentioned review article also has a discussion on bonding, so it can be cited in this context too. The authors refer to the application of elastic-plastic behaviour using Hill criterion in line 74. Here reference should be made to https://doi.org/10.3390/polym14194222, which gives both experimental and numerical analysis applying orthotropy combined with Hill criterion.  

Line 125: the authors use term 'three-point bending'. In fact, what they do is four-point bending.  

Lines 160-167, Table 1: One set of material constants are shown, while the cited sources (7,11,13) have different material properties. In the FEM analysis, the appropriate parameters should be used specifically for each case study.  

Fig.1. should be improved. Please create a high quality image (better resolution, proper font size, etc.).  

Stress diagrams in Figs. 6-8 are not ideal. The highest tensile stress range (gray colour) is too large and cannot show details. For example in Fig.6(b) the gray zone is between 40 and 227.4 and basically covers the entire lower half of the beam. I recommend that the authors may consider to apply a more uniform scaling if the software allows that so that a more detailed distribution is shown along the depth of the beam. Similar problems occur in Figs. 10-13. In comparison, the scale in Fig. 14. is good because it is more uniform and shows the details.  

After the parametric study, some comparison with literature results of similar studies might be added briefly.  

Section Conclusions is excessive. Basically it repeats all the results presented before. The purpose of Conclusions is to summarize the main conclusions drawn from the investigation. The authors may consider to rework it and reduce its length.

Author Response

Manuscript ID: materials-2810909

Improving the flexural response of timber beams using externally bonded carbon fibre-reinforced polymer (CFRP) sheets

Dear Editor,

The authors appreciate the suggestions of the reviewer aiming at enhancing the quality of the paper. The valuable comments have been considered in the revised manuscript as elaborated below:

Reviewer (2):

  • The title is too general. A more specific title to fit the contents of the work might be considered.
  •  The title has been edited.
  • The literature survey has to be extended with a few relevant items of recent years as follows. A review article on reinforcing timber with FRP is given by https://doi.org/10.3390/polym14122381, which should be cited. Another review is given by http://dx.doi.org/10.32604/jrm.2022.021983. The authors address bonding in line 59. The first mentioned review article also has a discussion on bonding, so it can be cited in this context too. The authors refer to the application of elastic-plastic behaviour using Hill criterion in line 74. Here reference should be made to https://doi.org/10.3390/polym14194222, which gives both experimental and numerical analysis applying orthotropy combined with Hill criterion.   
  •  All mentioned references have been cited in the introduction as required.
  • Line 125: the authors use term 'three-point bending'. In fact, what they do is four-point bending.  
  •  Edited as required.
  • Lines 160-167, Table 1: One set of material constants are shown, while the cited sources (7,11,13) have different material properties. In the FEM analysis, the appropriate parameters should be used specifically for each case study.  
  •  Table 1 has been listed as per the recommendation of only Shekarchi et al. [20]. Such meaning has been edited in the manuscript.
  • 1. should be improved. Please create a high quality image (better resolution, proper font size, etc.).  
  •  High quality image has been used in Fig. 1.
  • Stress diagrams in Figs. 6-8 are not ideal. The highest tensile stress range (gray colour) is too large and cannot show details. For example in Fig.6(b) the gray zone is between 40 and 227.4 and basically covers the entire lower half of the beam. I recommend that the authors may consider to apply a more uniform scaling if the software allows that so that a more detailed distribution is shown along the depth of the beam. Similar problems occur in Figs. 10-13. In comparison, the scale in Fig. 14. is good because it is more uniform and shows the details.  
  •  This is the goal from this section to show that the stress limit for elastic model is too much higher than the plastic mode.
  • After the parametric study, some comparison with literature results of similar studies might be added briefly.  
  • Literature results have been added in the introduction not to increase the length of the paper.
  • Section Conclusions is excessive. Basically it repeats all the results presented before. The purpose of Conclusions is to summarize the main conclusions drawn from the investigation. The authors may consider to rework it and reduce its length
  • Conclusions section has been totally re-written as required.